# Distribution and Maturity of Medial Collagen Fibers in Thoracoabdominal Post-Dissection Aortic Aneurysms: A Comparative Study of Marfan and Non-Marfan Patients

**DOI:** 10.3390/ijms26010014

**Published:** 2024-12-24

**Authors:** Panagiotis Doukas, Bernhard Hruschka, Cathryn Bassett, Eva Miriam Buhl, Florian Simon, Pepijn Saraber, Michael Johan Jacobs, Christian Uhl, Leon J. Schurgers, Alexander Gombert

**Affiliations:** 1Department of Vascular Surgery, RWTH Aachen University Hospital, 52074 Aachen, Germany; behruschka@ukaachen.de (B.H.); cbassett@ukaachen.de (C.B.); mjacobs@ukaachen.de (M.J.J.); cuhl@ukaachen.de (C.U.); agombert@ukaachen.de (A.G.); 2Institute of Pathology, Electron Microscopy Facility, RWTH Aachen University Hospital, 52074 Aachen, Germany; 3Clinic for Vascular and Endovascular Surgery, University Hospital Duesseldorf, Heinrich-Heine-University Duesseldorf, 40225 Düsseldorf, Germany; florian.simon@med.uni-duesseldorf.de; 4Department of Biochemistry, Cardiovascular Research Institute Maastricht (CARIM), Maastricht University, 6200 MD Maastricht, The Netherlands; pepijn.saraber@maastrichtuniversity.nl (P.S.); l.schurgers@maastrichtuniversity.nl (L.J.S.)

**Keywords:** thoracoabdominal aortic aneurysms, Marfan syndrome, collagen fibers, extracellular matrix, aortic aneurysm, polarized light microscopy, electron microscopy

## Abstract

Thoracoabdominal aortic aneurysms (TAAAs) are rare but serious conditions characterized by dilation of the aorta characterized by remodeling of the vessel wall, with changes in the elastin and collagen content. Individuals with Marfan syndrome have a genetic predisposition for elastic fiber fragmentation and elastin degradation and are prone to early aneurysm formation and progression. Our objective was to analyze the medial collagen characteristics through histological, polarized light microscopy, and electron microscopy methods across the thoracic and abdominal aorta in twenty-five patients undergoing open surgical repair, including nine with Marfan syndrome. While age at surgery differed significantly between the groups, maximum aortic diameter and aneurysm extent did not. Collagen content increased from thoracic to infrarenal segments in both cohorts, with non-Marfan patients exhibiting higher collagen percentages, notably in the infrarenal aorta (729.3 nm vs. 1068.3 nm, *p* = 0.02). Both groups predominantly displayed mature collagen fibers, with the suprarenal segment containing the highest proportion of less mature fibers. Electron microscopy revealed comparable collagen fibril diameters across segments irrespective of Marfan status. Our findings underscore non-uniform histological patterns in TAAAs and suggest that ECM remodeling involves mature collagen deposition, albeit with lower collagen content observed in the infrarenal aorta of Marfan patients.

## 1. Introduction

Thoracoabdominal aortic aneurysms (TAAAs) are relatively rare forms of degenerative disease of the aortic wall characterized by the abnormal dilation of the aorta and are associated with the risk of rupture and subsequent mortality [1]. Aneurysm formation and expansion are often correlated with atherosclerosis and hypertonia among other cardiovascular risk factors [2]. Aortic dissections also influence the biomechanical properties of the aortic wall and can promote the development of TAAAs, necessitating regular monitoring to preemptively address aneurysm growth before reaching a critical, rupture-prone size [3,4].

A unique subgroup affected by these aneurysms includes individuals with Marfan syndrome, a genetic condition predisposing them to connective tissue abnormalities [5]. Marfan syndrome, an autosomal dominant disorder resulting from mutations in the *fibrillin-1 (FBN1)* gene, leads to systemic connective tissue weakness, predominantly affecting the cardiovascular, skeletal, and ocular system [6].

In contrast, TAAAs in non-Marfan syndrome patients develop primarily due to acquired factors such as chronic atherosclerosis, systemic hypertension, and age-related degeneration of the extracellular matrix (ECM) [2,3]. Unlike Marfan patients, where genetic defects predispose them to early elastin degradation, non-Marfan patients experience ECM remodeling as a secondary effect of prolonged exposure to cardiovascular risk factors.

However, the pathological mechanisms underlying aneurysm formation in Marfan and non-Marfan patients, as well as in an angiotensin II (AngII)-infused ApoE−/− mouse model of AAAs [7], are similar and involve alterations in the ECM composition and structure, particularly affecting elastin and collagen fibers [8], which play an important role in providing the aortic wall with elasticity and tensile strength [9]. These alterations contribute to increased aortic wall stiffness and susceptibility to aneurysmal changes [10]. Although changes in collagen content and structure have been documented in cases of isolated, infrarenal abdominal aneurysms (AAAs) and ascending thoracic aneurysms [11,12,13], data on the histopathological characteristics of the whole thoracoabdominal aorta are scarce.

Many studies focus on the content, biomechanical properties, and structure of adventitial collagen regarding aneurysm formation [8,14,15], since a significant amount collagen deposition takes place in the remodeling of the aneurysmatic abdominal aorta [8]. However, the medial layer of the aortic wall, which contains a high proportion of vascular smooth muscle cells and elastic fibers, also undergoes significant changes in AAAs and is often in the center of the remodeling processes in the aortic wall [16]. The medial layer’s collagen content is crucial for providing structural support and maintaining the aorta’s ability to withstand hemodynamic stress. In AAAs, there is a notable degradation of ECM components, including collagen, and while the ratio of collagen type I to type III in the medial layer of isolated aortic aneurysms reportedly remains unchanged (2:1) [17], changes in total collagen content are still a matter of discussion [11,14].

Given the genetic predisposition for elastin degradation in Marfan patients and the accompanying remodeling of the ECM, gaining insight on the degradation, synthesis, and structure of vascular collagen fibers is crucial for understanding the underlying mechanisms of aneurysm progression. This study aims to elucidate the differences in collagen fiber composition and structural integrity in the aortic segments of Marfan and non-Marfan patients. We hypothesize that the ECM in Marfan patients exhibits distinct collagen characteristics due to ECM remodeling under elastic fiber degradation. By employing polarized light microscopy and electron microscopy, we provide a comprehensive analysis of collagen fiber distribution, maturity, and morphological features across different aortic segments. This investigation seeks to enhance our understanding of the pathophysiological distinctions in aortic aneurysms between Marfan and non-Marfan populations, ultimately informing tailored therapeutic approaches and improving patient management strategies.

## 2. Results

Out of the twenty-five patients included in this cohort, seven (28%) were women, and nine (33.3%) had a prior diagnosis of Marfan syndrome (Table 1). Marfan patients were significantly younger at the time of operation compared to non-Marfan patients (42.56 ± 10.06 vs. 55.88 ± 8.45, *p* = 0.005). However, there was no significant difference in the maximum aortic diameter between the two groups. Additionally, the extent of the aneurysms, classified according to the Crawford classification, did not differ significantly between the groups.

Increased intima thickness was observed in the infrarenal aortic segment in both groups (Table 2, Figure 1). This intima hyperplasia was particularly pronounced in non-Marfan patients (1068.3 μm vs. 729.7 μm, *p* = 0.02).

### 2.1. Collagen Fiber Analysis

The percentage of collagen fibers in the aortic media increased progressively from the thoracic part to the infrarenal segment, with the infrarenal segment showing the highest values (29.7% in Marfan patients and 54.1% in non-Marfan patients) (Table 3). These differences in collagen percentage between segments were statistically significant for both groups (Marfan *p* = 0.002 and non-Marfan *p* < 0.001) (Figure 2). Women had significantly lower collagen content in the suprarenal and infrarenal aorta (Appendix A). Across all segments, Marfan patients exhibited lower percentages of medial collagen fibers compared to non-Marfan patients. Although the largest difference was observed in the infrarenal segment, this difference was not statistically significant (29.7% vs. 54.1%, *p* = 0.06). Women had significantly lower collagen percentages in the suprarenal and infrarenal segments (Table 4). Men and women had no significant age difference (men 50.1 ± 10.6 vs. women 49.4 ± 9.1 years, *p* = 0.9).

Collagen fibers were quantified using polarized light microscopy based on their maturity, ranging from red (mature fibers) to teal (less mature fibers) (Figure 3). Across all segments and both groups (Marfan and non-Marfan), the majority of collagen fibers were mature (red or orange) (Table 5, Figure 4). No significant differences were found between the two groups in any aortic segment. However, the highest percentage of less mature fibers (green or teal) was found in the suprarenal aortic segment (14.8% green, 16.4% teal), with this difference being statistically significant for the non-Marfan patients (*p* < 0.001 for both green and teal).

### 2.2. Electron Microscopy

The configuration of the collagen fibers in the aortic segments, as revealed by electron microscopy, showed no differences between Marfan and non-Marfan patients (Figure 5). The mean diameter of the fibers remained constant throughout the aortic segments in both groups (Table 6). The waviness of the collagen fibers was similar in every segment for both groups.

## 3. Discussion

Our study offers significant insights into the distribution and maturity of medial collagen fibers in the different aortic segments of TAAAs among Marfan and non-Marfan patients, describing an overall similar background of collagen characteristics within the two groups, but significant differences across the aortic segments.

The progressive increase in medial collagen content from the thoracic to the infrarenal aortic segments, as presented in this study, shows that the extent of ECM remodeling in response to aneurysm formation and expansion differs in the different aortic segments. Our findings contradict previous observations, which describe a decrease in total collagen from the descending to the infrarenal aorta [18]. Specifically, Halloran et al., in their analysis of nine non-aneurysmal aortas obtained post-mortem, observed a significant reduction in collagen content relative to luminal surface area in the infrarenal segment compared to the ascending aorta, with no significant differences between the thoracoabdominal segments. This discrepancy between studies may be attributed to differences in study design. Unlike the autopsy material used by Halloran et al., which may have been subject to post-mortem tissue degradation, our samples were harvested during surgery and immediately processed for histological analysis, preserving their structural integrity. The increased collagen content observed in the infrarenal aorta of our cohort aligns with findings in infrarenal aortic tissue of AAAs. While increased collagen deposition in isolated AAAs is a well-established phenomenon [17], our results suggest a similar remodeling pattern in the infrarenal segment of TAAAs. However, these findings should be interpreted within the context of our distinct and homogeneous cohort, which exclusively consists of patients suffering from post-dissection aneurysms.

A key finding of our study is the significant difference in the total collagen content in the infrarenal aorta between Marfan and non-Marfan patients, with non-Marfan patients exhibiting nearly double the collagen content in this segment. Collagen deposition compensatory to the loss of vascular smooth muscle cells has been previously described in the literature, and it increases with the progression of isolated infrarenal abdominal aneurysms [19,20]. The susceptibility of the infrarenal segment of the aorta to wall changes may be explained through the different embryologic background of the different aortic segments, resulting in a heterogenous composition of the thoracoabdominal aortic wall [21]. Furthermore, oxygen supply to the aortic smooth cells varies across the different segments, with the smooth muscle cells of the infrarenal aorta relying solely on transintimal perfusion, whereas the thoracic segment is presented with both vascular and avascular regions [22].

In our study cohort, women exhibited lower collagen percentages across all examined aortic segments. This observation aligns with previous research linking females to an increased risk of abdominal aortic aneurysm rupture [23], with reduced medial collagen content proposed as a contributing factor [24]. However, Villard et al. [25] challenged this hypothesis, reporting no significant sex-based differences in collagen composition but emphasizing variations in collagen cross-linking. Our investigation, adopting a distinct perspective and focusing on post-dissection TAAAs, supports the findings of Tong et al. [24] and extends the hypothesis of reduced collagen in women. In TAAAs, we observed a notable reduction in collagen across all aneurysmal segments.

A further factor correlating with the increase in collagen content is age [26], which is associated with increased density of cross-links in collagenous tissue and has a stiffening effect on the mechanical response of the aortic wall [27,28]. In our study, non-Marfan patients were significantly older than the Marfan group, which may partly explain the difference in collagen deposition, with a further possible explanation lying in inflammation. Inflammatory processes taking place in the infrarenal part of the aorta are in the center point of ECM remodeling in non-Marfan patients, leading to increased collagen deposition [29,30]. This process was less pronounced in patients with Marfan syndrome, as demonstrated in the less pronounced intima hyperplasia between the two groups.

Polarized light microscopy analysis revealed no significant differences in the maturity and morphological features of collagen fibers between Marfan and non-Marfan patients. Most collagen fibers were mature, suggesting that ECM remodeling in TAAAs predominantly involves deposition of mature collagen fibers, irrespective of the underlying genetic condition. However, the notable presence of less mature fibers in the suprarenal aortic segment indicates a dynamic remodeling process where new collagen fibers are continually being synthesized to maintain aortic wall integrity under mechanical stress, possibly due to the relative movement of the suprarenal aortic segment to the diaphragmatic crux.

Evaluating the diameter of medial collagen fibrils in Marfan and non-Marfan patients showed no significant differences. The thickest fibers on average were observed in the thoracic segment and the thinnest ones in the suprarenal segment. This finding is in accordance with the observations from the polarized light microscopy, where the less mature fibers were more common in the suprarenal segment than in other segments. However, the mean diameter in all segments was higher than the diameter previously described in isolated infrarenal AAAs [7], and the fibers did not present with an abnormal appearance, implying possible differences in collagen degradation processes between post-dissection aneurysms and sporadic, infrarenal AAAs or between fresh samples and obduction material.

Our study has several limitations that should be acknowledged. Given the considerable challenge of obtaining fresh samples of the thoracoabdominal aorta, healthy or non-aneurysmatic dissections, the results of this work can only describe the condition of the aortic wall in aneurysms with a treatment indication. The focus on histopathological characteristics without incorporating biomechanical analyses leaves gaps in understanding the functional implications of the observed differences in collagen content and structure. Furthermore, there was assessment of the quality of the collagen fibers and the changes in fibril curvature and depth of D-periods as reported in other studies [7]. Additionally, this investigation focused mainly on the content and maturity of collagen fibers of the media layer and did not include assessment of adventitial collagen fibers or other components of the media layer, such as vascular smooth muscle cells and elastin. While we discussed inflammation as a potential factor influencing collagen deposition, we did not directly measure inflammatory markers. Future studies should include these measurements to clarify the role of inflammation in ECM remodeling. Lastly, our study did not explore genetic or molecular differences beyond the diagnosis of Marfan syndrome. Detailed genetic and molecular analyses could uncover specific pathways involved in ECM remodeling and aneurysm formation in both patient groups.

## 4. Materials and Methods

### 4.1. Study Design and Patient Selection

This single-center, prospective study recruited 25 patients scheduled for elective open TAAA repair for post-dissection aneurysms between May 2021 and November 2022. All dissections were either chronic type B dissections or residual thoracoabdominal dissections of previously repaired type A dissections. The study protocol was reviewed and approved by the ethics committee of the University Hospital Aachen (EK102/20). All patients provided written informed consent prior to participation. Data on patients’ medical history and demographic details were collected from digital medical records and clinical charts. Marfan diagnosis was confirmed in prior genetic testing.

### 4.2. Tissue Sample Preparation and Histology

Fresh, intraoperative tissue samples were resected and processed in a sterile environment. Only surgical samples of the diseased segments were extracted, taking care to include all three layers of the aortic wall. Samples for histological examination were stored in a formaldehyde solution (3.5–4%). The samples were embedded in paraffin, and sections were cut at 2 micrometers using a Leica SM200R microtome (Leica Microsystems, Wetzlar, Germany). Samples were stained with Hematoxylin-Eosin (HE) (“Hämalaun” Merck, Darmstadt, Germany, Art-Nr.: 1.09249.2500, and “Eosin G” Roth-Chemie Karlsruhe, Germany, Art.- Nr. X883.2) and Picrosirius Red (“Pikro-Siriusrot für Kollagen I und III”, Morphisto GmbH, Offenbach, Germany, Art-Nr.: 13425). All HE and Picrosirius Red sections, as well as the immunohistological preparations, were scanned with a FRITZ Slidescanner (PreciPoint GmbH, Munich, Germany) at 20× magnification at the Interdisciplinary Center for Clinical Research (IZKF) of the University Hospital Aachen. The evaluation of Picrosirius Red sections using polarization microscopy was carried out in the Department for Pathology of the MUMC+ in Maastricht, Netherlands, using a Leica 5000b microscope (Leica Microsystems GmbH, Wetzlar, Germany) and custom software developed by Dr. ir. Jack Cleutjens (Department for Pathology, MUMC+ Maastricht, Netherlands). Picrosirius Red slides were imaged using circularly polarized light, showing thick, mature collagen fibers as red/orange and thin and less mature fibers as green/teal. Mature collagen fibers are defined as those that have undergone complete cross-linking, contributing to tensile strength and stability within the extracellular matrix. In contrast, immature collagen fibers are less cross-linked and represent newly synthesized fibers that are more susceptible to degradation and remodeling [31]. All samples were orientated in the same direction for polarized light microscopy. Total collagen content was analyzed using the open-source software QuPath (Open source software for digital pathology image analysis, Version 0.3.4). The media layer in each sample was marked, and a color threshold was set for the detection of collagen fibers, resulting in the percentage of collagen stain relative to the media layer. Artifacts and folds in the samples were manually excluded.

### 4.3. Electron Microscopy

Tissue samples from 3 Marfan and 3 non-Marfan patients were chosen for evaluation of the collagen fibril diameter with electron microscopy. Small pieces of aorta segments were fixed first in formalin and then in 3% glutaraldehyde in 0.1 M Soerensen’s phosphate buffer. Samples were treated with 0.25% tannic acid (Mallinckrodt, Paris, KY, USA) in 0.1 M Soerensen’s phosphate buffer for 24 h at room temperature, rinsed in Soerensen’s phosphate buffer, incubated in 1% OsO4 (Roth, Karlsruhe, Germany) in 25 mM sucrose buffer (Merck, Darmstadt, Germany), and then dehydrated via ascending ethanol series (30, 50, 70, 90, and 3 × 100%) for 10 min each. Dehydrated specimens were incubated in propylene oxide (Science Services, Munich, Germany) for 30 min, in a 1:1 mixture of Epon resin (Serva, Heidelberg, Germany) and propylene oxide for 1 h, and finally in pure Epon for 1 h. Epon polymerization was performed at 90 °C for 2 h. Ultrathin sections were stained with 0.5% uranyl acetate and 1% lead citrate (both Science Services, Munich, Germany) to enhance contrast. Samples were examined using a transmission electron microscope (Zeiss Leo906, Oberkochen, Germany) operating at an acceleration voltage of 60 kV. The diameter of individual fibers was measured at the points of clear contours, and a mean was calculated for at least 10 measurements per aortic segment. An example of the measurements is displayed in Appendix A.

### 4.4. Data Analysis

Continuous variables are presented as mean ± standard deviation or median [IQR] in cases of skewed distributions. Categorical data are presented as absolute counts and percentages. Missing values were addressed using a multiple imputation by chained equations (MICEs). Statistical significance for repeated measurements was tested with Friedman’s ANOVA, followed by Dunn-Sidàk post hoc test. For independent variables, the Mann–Whitney test was used. If data were normally distributed, as assessed from Kolmogorov–Smirnov tests, Student’s t-test was employed. Statistical analysis and graph creation were performed using GraphPad Prism version 8.0.0 for Windows (GraphPad Software, San Diego, CA, USA).

## 5. Conclusions

In conclusion, our findings show that TAAAs display histopathological differences across the affected aortic segments which are not entirely uniform in their histology. ECM remodeling differs across the aortic segments, with the suprarenal aortic segment showing the highest percentage of newly synthesized collagen. Marfan and non-Marfan patients show no significant differences in size and maturity of collagen fibers, except for the infrarenal aorta, where Marfan patients have a significantly lower content of collagen than non-Marfan patients. Our results can guide future trials to investigate dysfunctional matrix turnover in pathologies extending in the thoracic and abdominal aorta in patients with and without connective tissue diseases.

## Figures and Tables

**Figure 1 ijms-26-00014-f001:**
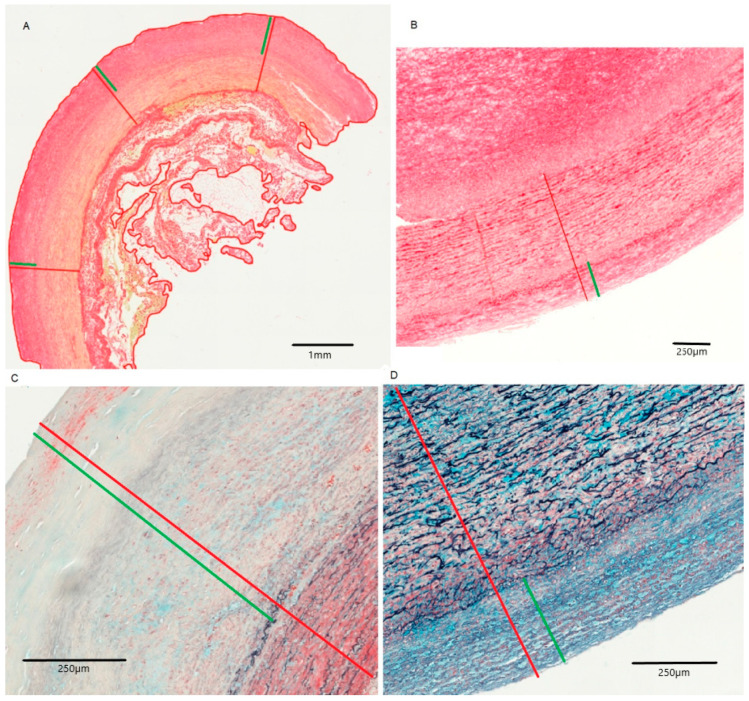
Intima and media of the infrarenal aorta for non-Marfan (**A**,**C**) and Marfan (**B**,**D**) patients. (**A**,**B**): picrosirius red staining, (**C**,**D**): MOVAT pentachrome staining. Red: intima and media, Green: only intima.

**Figure 2 ijms-26-00014-f002:**
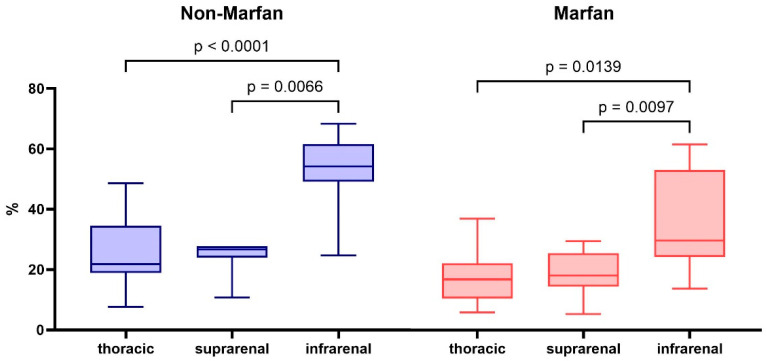
Total collagen percentage in the aortic media of the different aortic segments for non-Marfan and Marfan patients. *p*-values calculated with the Dunn-Sidàk post hoc test after Friedman ANOVA.

**Figure 3 ijms-26-00014-f003:**
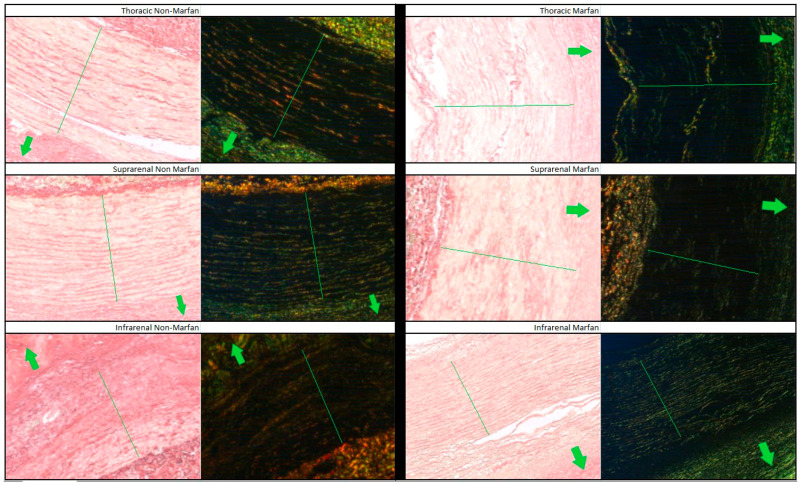
Picrosirius red staining of the different aortic segments for Marfan and non-Marfan patients. The images on the left were taken with conventional microscopy, while those on the right were taken with polarized light microscopy. The media layer is indicated by the green line running across the diameter. Arrows point to the aortic lumen. 40× magnification.

**Figure 4 ijms-26-00014-f004:**
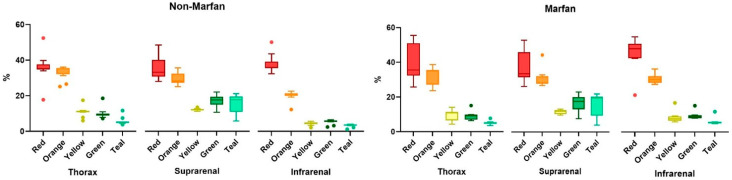
Quantification of the collagen fiber types in aortic media, according to level of maturity (red to green—mature to less mature).

**Figure 5 ijms-26-00014-f005:**
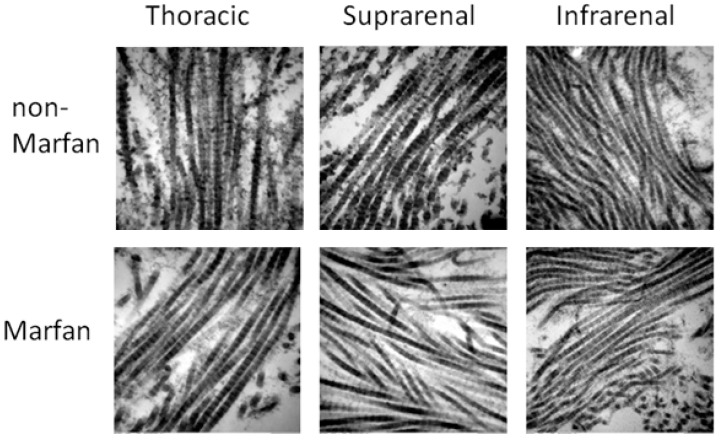
Collagen fibers in the different aortic segments. 60,000× magnification.

**Table 1 ijms-26-00014-t001:** Demographics and baseline characteristics. Aneurysm extent is reported according to the Crawford classification. Statistical significance for *p* < 0.05 is marked with *.

	All (n = 25)	Marfan (n = 9)	Non-Marfan (n = 16)	*p*-Value
Age (years)	51.1 ± 11	42.56 ± 10.06	55.88 ± 8.45	0.005 *
Women	7 (28)	4 (44.4)	3 (18.8)	0.2
Men	18 (72)	5 (55.6)	13 (81.2)	0.2
Maximal aortic diameter (mm)	60.53 ± 14.15	58.81 ± 11.45	61.49 ± 15.74	0.63
Aneurysm extent				0.56
Crawford I	4 (16)	1 (11.1)	3 (18.8)	
Crawford II	10 (40)	5 (55.6)	5 (31.3)	
Crawford III	5 (20)	2 (22.2)	3 (18.8)	
Crawford IV	6 (24)	1 (11.1)	5 (31.3)	

**Table 2 ijms-26-00014-t002:** Thickness of the intima layer (μm). Statistical significance for *p* < 0.05 is marked with *.

	Descending Thoracic Aorta	Suprarenal Aorta	Infrarenal Aorta	Friedman *p*-Value
Marfan	318.53 [242.4–426.4]	390.03 [356.3–407.5]	729.72 [341.6–1032.4]	0
Non-Marfan	568.94 [436.9–602.8]	414.9 [379–421.9]	1068.30 [1036.8–1147.7]	<0.001 *
Mann–Whitney *p*-value	0.03 *	0.14	0.02 *	

**Table 3 ijms-26-00014-t003:** Percentage of total collagen in the aortic media. Statistical significance for *p* < 0.05 is marked with *.

	Descending Thoracic Aorta	Suprarenal Aorta	Infrarenal Aorta	Friedman ANOVA *p*-Value
Marfan	16.8 [13.9–21.7]	18.1 [16.9–21]	29.7 [26.9–39]	0.002 *
Non-Marfan	21.9 [19.5–29.8]	26.7 [24.8–27.8]	54.1 [49.5–61.6]	<0.001 *
Mann–Whitney *p*-value	0.16	0.19	0.06	

**Table 4 ijms-26-00014-t004:** Percentage of total collagen in the aortic media of men and women. Statistical significance for *p* < 0.05 is marked with *.

	Descending Thoracic Aorta	Suprarenal Aorta	Infrarenal Aorta
Male	22.15 [18.18–32.16]	27.76 [20.84–27.76]	56.19 [40.03–61.63]
Female	15.89 [8.65–20.81]	18.14 [13.16–18.78]	28.17 [22.93–31.88]
*p*-value	0.086	0.037 *	0.014 *

**Table 5 ijms-26-00014-t005:** Collagen fiber maturity in the aortic media. Percentages of fibers categorized by color using polarized light microscopy. Statistical significance for *p* < 0.05 is marked with *.

		Descending Thoracic Aorta	Suprarenal Aorta	Infrarenal Aorta	Friedman *p*-Value
Red	Marfan	35.6 [34–41.9]	28.3 [26.7–35.3]	46 [41–48.5]	0.02
	Non-Marfan	35.3 [34.9–36.8]	28.6 [26.4–34.3]	48.2 [47.8–50.3]	<0.001 *
	*p*-value	0.49	0.86	0.08	
Orange	Marfan	35.3 [31.4–35.4]	23.7 [23.6–25.8]	28.9 [28.0–30.2]	0
	Non-Marfan	35.2 [33.9–35.5]	24.3 [23.6–27.5]	28.4 [27.4–28.5]	0.004 *
	*p*-value	0.749	0.799	0.151	
Yellow	Marfan	10.9 [9.2–11.4]	10.0 [9.1–10.5]	7.3 [6.2–7.9]	0.2
	Non-Marfan	11.3 [11.0–11.4]	10.6 [10.2–10.7]	6.3 [5.7–6.4]	<0.001 *
	*p*-value	0.4	0.1	0.104	
Green	Marfan	9.1 [8.0–9.7]	14.8 [12.3–16.8]	8.3 [7.7–8.9]	0.09
	Non-Marfan	9.6 [9.3–9.8]	15.2 [13.0–16.7]	7.9 [7.7–8.0]	<0.001 *
	*p*-value	0.551	0.754	0.09	
Teal	Marfan	4.9 [4.6–5.2]	16.4 [10.0–16.9]	5.3 [4.8–5.5]	0.09
	Non-Marfan	5.0 [4.9–5.2]	15.3 [10.5–16.8]	4.8 [4.6–5.0]	<0.001 *
	*p*-value	0.47	0.96	0.13	

**Table 6 ijms-26-00014-t006:** Mean diameter of the collagen fibers (nm).

	Descending Thoracic Aorta	Suprarenal Aorta	Infrarenal Aorta
Marfan	63.4 ± 8.4	50.2 ± 4.4	59.8 ± 3.5
Non-Marfan	64.4 ± 1	58.4 ± 14	54 ± 8.6
*p*-value	0.56	0.51	0.45

## Data Availability

The original contributions presented in the study are included in the article/Appendix A, further inquiries can be directed to the corresponding author.

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
