# Peer review of "Distribution and Maturity of Medial Collagen Fibers in Thoracoabdominal Post-Dissection Aortic Aneurysms: A Comparative Study of Marfan and Non-Marfan Patients"

_ijms, 2024, doi:10.3390/ijms26010014_

Round 1
Reviewer 1 Report (Previous Reviewer 2)
Comments and Suggestions for Authors
I have nothing further
Author Response
Dear Reviewer,
Thank you for your positive feedback. We are pleased that the revised version of our manuscript sufficiently addresses your concerns.
Reviewer 2 Report (New Reviewer)
Comments and Suggestions for Authors
Reviewing the review manuscript entitled, “Distribution and Maturity of Medial Collagen Fibers in Thoracoabdominal Post-Dissection Aortic Aneurysms: A Comparative Study of Marfan and Non-Marfan Patients” by Doukas P et al., this is an article focusing on differences in the distribution of collagen fibers in TAAAs between marfan and non-marfan patients. I highly appreciate that this is a prospective study involving patients with aortic dissection but there are some concerns. The authors should address the following concerns.
The authors should describe the mechanism of development of non-Marfan syndrome patients in the introduction. If the authors classify them into two groups, you should provide a clear comparison in a table.
The authors mentioned “This study aims to elucidate the differences in collagen fiber composition and structural integrity in the aortic segments of Marfan and non-Marfan patients.”. What clinical advantages does this objective provide for each disease?
The authors also mentioned “Across all segments, Marfan patients had consistently lower percentages of medial collagen fibers compared to non-Marfan patients, with the most notable, albeit not statistically significant, difference observed in the infrarenal segment. This is clearly an exaggeration. The authors should change the description to reflect the statistical results.
In Table 1, the authors should include male as an independent factor. There is no explanation of the Crawford classification.
The authors should provide a definition of mature and immature fibers.
The authors should modify Figure 3. I cannot understand the difference between the two conditions such as Marfan or non-Marfan.
In the discussion section, the authors mentioned “Our findings contradict previous observations, which describe a decrease in total collagen from the descending to the infrarenal aorta.”. Indeed, the authors opined that this was a prospective study and samples were obtained intraoperatively. I accept your opinions, but if you obtain results that differ from previous reports, you should justify your results in light of previous manuscripts related to your research.
Author Response
Reviewing the review manuscript entitled, “Distribution and Maturity of Medial Collagen Fibers in Thoracoabdominal Post-Dissection Aortic Aneurysms: A Comparative Study of Marfan and Non-Marfan Patients” by Doukas P et al., this is an article focusing on differences in the distribution of collagen fibers in TAAAs between marfan and non-marfan patients. I highly appreciate that this is a prospective study involving patients with aortic dissection but there are some concerns. The authors should address the following concerns.
Comment 1: The authors should describe the mechanism of development of non-Marfan syndrome patients in the introduction. If the authors classify them into two groups, you should provide a clear comparison in a table.
Response 1: Thank you for this suggestion. We have revised the introduction to specifically address the interplay of acquired factors that may lead to TAAA formation in non-Marfan syndrome patients, as outlined in lines 47–51:
“In contrast, TAAAs in non-Marfan syndrome patients develop primarily due to acquired factors such as chronic atherosclerosis, systemic hypertension, and age-related degeneration of the extracellular matrix (ECM). Unlike Marfan patients, where genetic defects predispose to early elastin degradation, non-Marfan patients experience ECM remodeling as a secondary effect of prolonged exposure to cardiovascular risk factors.”
While we appreciate your suggestion to include a table comparing these mechanisms, we believe the textual description now sufficiently addresses the differences and provides the necessary context for the study. Including a table may risk oversimplifying the complex interplay of genetic and acquired factors, which we feel is better conveyed in the revised narrative. We hope this revision meets your expectations, but we remain open to further refinements if you feel additional clarification is needed.
Comment 2: The authors mentioned “This study aims to elucidate the differences in collagen fiber composition and structural integrity in the aortic segments of Marfan and non-Marfan patients.”. What clinical advantages does this objective provide for each disease?
Response 2: While this study does not offer direct clinical applications, it provides a foundation for advancing the understanding of the pathomechanisms underlying TAAA, which could inform future research aimed at improving clinical management strategies. Understanding the unique patterns of ECM remodeling and collagen degradation in Marfan and non-Marfan patients may help identify specific biomarkers or structural vulnerabilities associated with aneurysm progression. These insights could enable earlier detection of high-risk patients through targeted imaging or biomarker-based screening, facilitating more precise timing for surgical interventions. By enhancing our understanding of the differences in collagen fiber composition and structural integrity between Marfan and non-Marfan populations, this study lays the groundwork for more tailored therapeutic approaches and ultimately improved patient outcomes for both groups.
Comment 3: The authors also mentioned “Across all segments, Marfan patients had consistently lower percentages of medial collagen fibers compared to non-Marfan patients, with the most notable, albeit not statistically significant, difference observed in the infrarenal segment. This is clearly an exaggeration. The authors should change the description to reflect the statistical results.
Response 3: Thank you for bringing this to our attention. We changed the wording as suggested.
Lines 110-113:
“Across all segments, Marfan patients exhibited lower percentages of medial collagen fibers compared to non-Marfan patients. Although the largest difference was observed in the infrarenal segment, this difference was not statistically significant.”
Comment 4: In Table 1, the authors should include male as an independent factor. There is no explanation of the Crawford classification.
Response 4: Thank you for this comment. As suggested, we have included "Men" as an independent factor in Table 1. Additionally, we have clarified that the Crawford classification is used to report the extent of the aneurysm.
Comment 5: The authors should provide a definition of mature and immature fibers.
Response 5: Thank you for this comment. We added this information in lines 267-271:
“Mature collagen fibers are defined as those that have undergone complete cross-linking, contributing to tensile strength and stability within the extracellular matrix. In contrast, immature collagen fibers are less cross-linked and represent newly synthesized fibers that are more susceptible to degradation and remodeling.”
Comment 6: The authors should modify Figure 3. I cannot understand the difference between the two conditions such as Marfan or non-Marfan.
Response 6: Thank you for your valuable feedback. Figure 3 is intended to clarify the area of interest analyzed using polarized light microscopy. We acknowledge that the tunica media is overshadowed by the collagen-rich adventitia, making the differences between Marfan and non-Marfan conditions less directly discernible. To address this, we have revised the figure to focus specifically on the media, improving clarity. However, we note that the differences remain subtle, as described in Table 3.
Comment 7: In the discussion section, the authors mentioned “Our findings contradict previous observations, which describe a decrease in total collagen from the descending to the infrarenal aorta.”. Indeed, the authors opined that this was a prospective study and samples were obtained intraoperatively. I accept your opinions, but if you obtain results that differ from previous reports, you should justify your results in light of previous manuscripts related to your research.
Response 7: Thank you for this important suggestion. We have revised the discussion section to provide a clearer contextualization of our findings within the existing body of evidence. The updated text is included in lines 163–176:
"Specifically, Halloran et al., in their analysis of nine non-aneurysmal aortas obtained post-mortem, observed a significant reduction in collagen content relative to luminal surface area in the infrarenal segment compared to the ascending aorta, with no significant differences between the thoracoabdominal segments. This discrepancy between studies may be attributed to differences in study design. Unlike the autopsy material used by Halloran et al., which may have been subject to post-mortem tissue degradation, our samples were harvested during surgery and immediately processed for histological analysis, preserving their structural integrity. Additionally, the increased collagen content observed in the infrarenal aorta of our cohort aligns with findings in infrarenal aortic tissue of abdominal aortic aneurysms (AAAs). While increased collagen deposition in isolated AAAs is a well-established phenomenon [17], our results suggest a similar remodeling pattern in the infrarenal segment of TAAAs. However, these findings should be interpreted within the context of our distinct and homogeneous cohort, which exclusively consists of patients suffering from post-dissection aneurysms."
Round 2
Reviewer 2 Report (New Reviewer)
Comments and Suggestions for Authors
This revised version reaches to an acceptable quality. Congrats.
This manuscript is a resubmission of an earlier submission. The following is a list of the peer review reports and author responses from that submission.
Round 1
Reviewer 1 Report
Comments and Suggestions for Authors
Well done on an elegant study that presents important results in the field of aortic research. However, I have a number of concerns.
This study focused solely on post-dissection aneurysm formation and, therefore, does not present an idea of what the cell content and structure of the aortic wall is normally like, i.e. how many of these changes result from a response to injury?
We need more information on the age of the dissection, i.e. are these acute or chronic changes in the wall?
Were these all primary type b dissections, or were some previously repaired type A dissections with residual thoracoabdominal dissection?
I note that the most significant difference between Marfan and non-Marfan patients is in the infrarenal aorta, whereas most Marfan-related dissection occurs in the ascending aorta. Why do the authors hypothesise that the differences are being expressed in the infrarenal aorta?
Did the authors consider differences in gender and relative collagen content? Gender is an important consideration in aortic disease since, although women are less often affected by aortic disease, the outcomes for female patients are worse.
I am not sure that the authors' conclusion that the higher content of immature collagen fibres in the suprarenal aorta is due to "hemodynamic stress in this region of the aorta is accurate. Indeed, the more proximal regions of the aorta are subject to higher hemodynamic stresses. Does the relative movement between the aorta and the diaphragmatic crux contribute to exposing the aorta to repetitive injury in this region?
In line 168, the authors refer to an inflammatory process in the ECM of non-Marfan patients that contributes to increased collagen content. However, surely if the Marfans patients have also been victims of dissection, they do have inflammatory processes at play? This is another reason why telling us the time since dissection is important.
Were the aortic wall samples taken from true or false lumen, or was this recorded? This should be uniform across all groups.
Although acknowledged by the authors, it is a shame that the adventitial layer was not examined, nor were other components of the aortic wall considered. Are there tissue samples fixed that could be examined for these factors?
Reviewer 2 Report
Comments and Suggestions for Authors
Collagen content in TAAA tissue
An inventory of the collagen content in the aorta of Marfan en non-Marfan patients is of interest, since aortic dissection is likely correlated to this feature, as vascular Ehlers Danlos syndrome with collagen mutations shows that an impaired collagen network is fundamental in arterial rupture (without aneurysm formation first). Quantity and Quality of collagen is different over the aortic trajectory and could provide insight in site specific ECM networks. While a reduction in elastin content in the aortic media is known from thoracic to abdominal, here an increase in collagen content is reported.
Among the non-Marfan patients were these patients all non-genetic aortic dissection cases or has there been established (later) that a genetic defect is present? They are quite young to have a dissection just because of age.
Since some of the tissues seem to have extensive intimal layers, which I am not used to for Marfan patients, it would be helpful to have elastin stainings to show where the boundary of the intima / media /adventitia is, because all data are based on the separation of the layers.
Under polarized light, my understanding is that the fiber orientation matters, meaning that it matters if all samples are oriented in the same way when measured. For example that the aortic tissue is always under the microscope with intima on top, media in the middle and adventitia on the bottom (and not from left to right). Since the aortic pictures under polarized light are of different orientations, I wonder if the same values of teal to red come out if the tissue is measured all in the same orientation. Please verify this.
There are various ECM quantification tools these days to assess more in depth the level of fibrosis beyond % of area. For example these two manuscripts: Digital Image Analysis of Picrosirius Red Staining: A Robust Method for Multi-Organ Fibrosis Quantification and Characterization. doi: 10.3390/biom10111585 or A FIJI macro for quantifying pattern in extracellular matrix. doi: 10.26508/lsa.202000880. This may provide more detailed insight?
The reduced collagen content in Marfan aortas is an important finding and likely contributing to reduced tensile strength and thus being more rupture prone. Could there be age matching for let’s say 3-5 patients and then see if the non-Marfan still have enhanced collagen content, just as proof of concept, to have an idea if it is related to age or Marfan?
